# Transcriptomic Profiling of DNA Damage Response in Patient-Derived Glioblastoma Cells before and after Radiation and Temozolomide Treatment

**DOI:** 10.3390/cells11071215

**Published:** 2022-04-04

**Authors:** Mathew Lozinski, Nikola A. Bowden, Moira C. Graves, Michael Fay, Bryan W. Day, Brett W. Stringer, Paul A. Tooney

**Affiliations:** 1School of Biomedical Sciences and Pharmacy, College of Health, Medicine and Wellbeing, University of Newcastle, Newcastle, NSW 2308, Australia; mathew.lozinski@uon.edu.au; 2Centre for Drug Repurposing and Medicines Research, University of Newcastle, Newcastle, NSW 2308, Australia; nikola.bowden@newcastle.edu.au (N.A.B.); moira.graves@newcastle.edu.au (M.C.G.); mike@advancell.com.au (M.F.); 3Hunter Medical Research Institute, New Lambton Heights, NSW 2305, Australia; 4School of Medicine and Public Health, College of Health, Medicine and Wellbeing, University of Newcastle, Newcastle, NSW 2308, Australia; 5GenesisCare, Newcastle, NSW 2290, Australia; 6QIMR Berghofer Medical Research Institute, Brisbane, QLD 4006, Australia; bryan.day@qimrberghofer.edu.au; 7College of Medicine and Public Health, Flinders University, Adelaide, SA 5042, Australia; brett.stringer@flinders.edu.au

**Keywords:** glioblastoma, DNA damage response, treatment resistance, temozolomide, radiation

## Abstract

Glioblastoma is a highly aggressive, invasive and treatment-resistant tumour. The DNA damage response (DDR) provides tumour cells with enhanced ability to activate cell cycle arrest and repair treatment-induced DNA damage. We studied the expression of DDR, its relationship with standard treatment response and patient survival, and its activation after treatment. The transcriptomic profile of DDR pathways was characterised within a cohort of isocitrate dehydrogenase (IDH) wild-type glioblastoma from The Cancer Genome Atlas (TCGA) and 12 patient-derived glioblastoma cell lines. The relationship between DDR expression and patient survival and cell line response to temozolomide (TMZ) or radiation therapy (RT) was assessed. Finally, the expression of 84 DDR genes was examined in glioblastoma cells treated with TMZ and/or RT. Although distinct DDR cluster groups were apparent in the TCGA cohort and cell lines, no significant differences in OS and treatment response were observed. At the gene level, the high expression of *ATP23*, *RAD51C* and *RPA3* independently associated with poor prognosis in glioblastoma patients. Finally, we observed a substantial upregulation of DDR genes after treatment with TMZ and/or RT, particularly in RT-treated glioblastoma cells, peaking within 24 h after treatment. Our results confirm the potential influence of DDR genes in patient outcome. The observation of DDR genes in response to TMZ and RT gives insight into the global response of DDR pathways after adjuvant treatment in glioblastoma, which may have utility in determining DDR targets for inhibition.

## 1. Introduction

Glioblastoma is the most common and aggressive type of primary malignant brain tumour in adults. The diagnosis of glioblastoma, updated in the WHO Classification of Tumors of the Central Nervous System 2021, involves a combination of histological (microvascular proliferation or necrosis) and molecular characteristics, including the criteria of having the isocitrate dehydrogenase (IDH) wild-type gene [1]. Standard treatment involves safe maximal surgical resection of the tumour; however, since glioblastoma has an indistinct tumour border, complete resection is not usually possible. Consequently, patients undergo intensive radiation therapy (RT) and temozolomide (TMZ) chemotherapy to treat residual tumour cells. These treatments cause single-stranded breaks (SSBs) or double-stranded breaks (DSBs) in the DNA of glioblastoma cells that may lead to cell cycle arrest and activation of cell death pathways [2]. Despite intense treatment, resistance frequently develops, causing a rapid recurrence at the primary tumour site, leaving patients with few treatment options, a poor prognosis and a median survival time of 15 months [3].

Evidence suggests that DNA damage response (DDR) pathways respond to treatment-induced damage and aid in tumour survival, leading to treatment resistance in glioblastoma [4]. The base excision repair (BER) pathway repairs SSBs caused by RT and methylated lesions caused by TMZ, whilst homologous recombination (HR), non-homologous end joining (NHEJ), and the Fanconi anemia (FA) pathway collectively respond to the potently cytotoxic DSBs and stalled replication forks caused by both treatment approaches. Furthermore, DDR is constitutively active as a consequence of oncogenic-induced replication stress in glioblastoma [5,6], with increased expression in a number of DDR pathways shown to facilitate treatment resistance [7,8]. Recent studies have profiled DNA repair pathways in glioblastoma to gain targetable insights in efforts to sensitise tumour cells to DNA-damaging agents [9,10].

We investigated the transcriptomic expression of DDR genes and pathways in a TCGA cohort of IDH wild-type glioblastoma patients and its relation to overall patient survival. We identified DDR profiles of 12 patient-derived glioblastoma cell lines and compared TMZ and RT sensitivity of cells stratified on the degree of DDR pathway expression. Lastly, these cell lines were exposed to a clinically relevant dose of RT and/or TMZ to investigate the extent and timing of DDR genes and pathways responding to treatment-induced damage.

## 2. Materials and Methods

### 2.1. Ethics

This study was approved by the Human Research Ethics Committee of the University of Newcastle (H-2020-0389).

### 2.2. TCGA Glioblastoma Cohort

To study the baseline DDR profile in glioblastoma, fragments per kilobase million (FPKM) values were collated from 140 IDH wild-type glioblastoma patients within the TCGA database [11]. FPKM values were converted to transcript per kilobase million (TPM) values [12] and single-sample gene set enrichment analysis (ssGSEA) was performed on the TCGA data in R. Gene sets were assigned corresponding to the major DDR pathways including BER, NER, mismatch repair (MMR), HR and NHEJ (Appendix A). Enrichment scores for each pathway were converted to log-transformed *z*-scores for data visualisation. RNAseq by expected maximisation (RSEM) data were collated for 84 DDR genes (Appendix A). Patient groups were stratified into “high” and “low” expression based on a median split of RSEM expression values per gene. The log-rank test was used to find potential differences in overall survival (OS) between groups. Multiple Cox regression was implemented on genes with significant difference in OS from the log-rank test (*p* < 0.05), using clinical covariables that were determined to be significant through univariate Cox regression (Appendix A).

### 2.3. Cell Lines and Reagents

Twelve patient-derived glioblastoma cell lines were kindly provided by the QIMR Berghofer Medical Research Institute (Brisbane, Australia). The cell lines are fully characterised with publicly available molecular and patient data, published by Stringer et al. [13]. Cells were grown as adherent monolayers in Matrigel^®^ (Corning^®^, Corning, NY, USA)-coated tissue culture flasks in StemPro^®^ NSC SFM (Gibco^TM^, Waltham, MA, USA) containing 100 I.U./mL penicillin and 100 µg/mL streptomycin (Gibco^TM^, Waltham, MA, USA), and incubated at 37 °C in 5% CO_2_/95% humified air. Cells were passaged using StemPro^®^ Accutase^®^ solution (Gibco^TM^, Waltham, MA, USA) for detachment of adherent cells. TMZ was purchased from Sigma-Aldrich (Burlington, MA, USA), aliquoted in dimethyl sulfoxide (DMSO) (100 mM) and stored between 2 and 8 °C. RT was delivered using a medical linear accelerator (LINAC) at GenesisCare, Gateshead NSW (Australia) or RS-2000 Small Animal Irradiator (Rad Source, Buford, GA, USA).

### 2.4. RNA Sequencing Analysis

RNA sequencing data from all 12 glioblastoma cell lines were obtained from the publicly available QMIR database (https://www.qimrberghofer.edu.au/commercial-collaborations/partner-with-us/q-cell/ accessed on 10 May 2021). RNA extraction methods and RNA sequencing analysis from the cell lines are described in Stringer et al. [13]. ssGSEA was performed on TPM values for each cell line, using the same gene sets as used in the TCGA cohort (Appendix A).

### 2.5. Cell Viability Assay

To assess cell viability, 96-well plates were coated with Matrigel under ice-cold conditions prior to plating with cells. Adherent glioblastoma cell lines were passaged and seeded at 4000 Cells/well in 96-well plates overnight, before treatment with a clinically relevant dose of TMZ (35 µM) [14] or RT (2 Gy). After 7 days, 50 µL of a 2 mg/mL solution of MTT Formazan (Sigma-Aldrich, USA) and Dulbecco’s Phosphate Buffer Saline (DPBS, without magnesium chloride and calcium chloride Gibco^TM^, Waltham, MA, USA) was added to each well and incubated for 3 h. The medium/MTT solution was aspirated and DMSO (120 µL) added into each well. Each plate was shaken using an IKA^®^ MS 3 basic shaker (Sigma Aldrich, Saint Louis, MO, USA) at 600 rpm for 2 min. Absorbance was read at 570 nM using the SPECTROstar^®Nano^ microplate reader (BMG LABTECH, Ortenberg, Germany).

### 2.6. Comparison of Cell Line DDR Profile and Treatment Response

Glioblastoma cell lines were stratified into two cluster groups (C1 and C2) based on hierarchical clustering of ssGSEA scores in each DDR pathway. Cell lines were also assigned into “high” or “low” expression groups according to each DDR pathway based on a median split of ssGSEA scores. Differences in TMZ or RT cell viability between cluster groups or “high” vs. “low” DDR expression groups was assessed through an unpaired student’s *t*-test.

### 2.7. Time Course and Quantitative PCR

Four glioblastoma cell lines (HW1, FPW1, SB2b and MN1) were seeded at 300,000 cells/well in 6-well Matrigel-coated plates overnight and treated with a clinically relevant dose of RT (2Gy), followed by TMZ (35 µM) one hour later. Cells were harvested at 2, 24 and 48 h after TMZ treatment and extracted for RNA using the AllPrep DNA/RNA/Protein Mini Kit (Qiagen, Germantown, MD, USA). RNA was converted to cDNA using the High-Capacity cDNA Reverse Transcription Kit (ThermoFisher Scientific, Waltham, MA, USA). In accordance with supplier instructions, gene expression of 84 DDR genes (Appendix A) was examined using a TaqMan™ ^®^Gene Expression Custom Array Card (ThermoFisher Scientific, Waltham, MA, USA). Samples were run as biological triplicates using the QuantStudio™ 7 Pro Real-Time PCR System (ThermoFisher Scientific, Waltham, MA, USA). The geometric means of housekeeping genes (Appendix A) were used to determine the absolute expression and fold changes of target genes for each cell line using the ΔCt and ΔΔCt method. Differentially expressed genes (DEGs) were identified within each cell line as significantly expressed genes for a particular treatment and time point when compared to the untreated control, using Dunnett’s multiple comparison test of absolute expression values [15].

### 2.8. Statistical Analysis

The statistical analyses including the Mann–Whitney test, the Kruskal–Wallis test and Dunnett’s multiple comparison test were conducted in GraphPad Prism 7. Unsupervised hierarchical clustering (Ward’s method) of ssGSEA scores was performed in R using the ‘stats’ package and visualised using the ‘ComplexHeatmap’ package. Survival analyses using the log-rank test and Cox regression were performed in R using the ‘survival’ package. *p*-values < 0.05 were considered significant.

## 3. Results

### 3.1. Expression of DDR Genes and Association with Patient Survival

First, we analysed RNA sequencing data from 140 IDH wild-type glioblastoma patient samples in the TCGA to determine distinct DDR profiles using ssGSEA. This method calculates enrichment scores of gene sets within a single sample and thus represents the degree to which such gene sets are up- or downregulated within a sample [16]. Five gene sets were used in this study, representing the five canonical DDR pathways (BER, MMR, NER, HR and NHEJ) [17]. Hierarchical clustering of ssGSEA scores from each DDR pathway identified three distinct clusters (TC1–TC3), wherein TC3 had the highest gene expression of each DDR pathway, followed by TC2, and lastly TC1, which had the lowest gene expression in each DDR pathway (Figure 1A). There was a trend of small increases in the proportion of *MGMT* methylated patients when comparing cluster C1 through to C3 (C1 = 34.2% (13/48), C2 = 41.5% (17/41), C3 = 53.3% (16/30)). A similar trend occurred for the proportion of *TP53* alterations (C1 = 13% (7/54), C2 = 20.8% (11/53), C3 = 33.3% (11/33)). Although distinct DDR gene profiles were apparent, no survival difference was found across clusters (Figure 1B).

Next, using the same TCGA patient cohort, we asked whether the expression of individual DDR genes could predict OS outcomes of glioblastoma patients. “High” and “low” expression groups were determined for DDR genes and Kaplan–Meier survival analysis was performed. After accounting for covariates using Cox regression (Appendix A), high expression of DDR genes *ATP23*, *RAD51C* and *RPA3* was independently associated with poorer overall patient survival (Figure 2).

### 3.2. Expression of DDR Pathways Influence Treatment Response in Glioblastoma Cell Lines

Next, we examined the baseline gene expression profiles of DDR pathways in 12 patient-derived glioblastoma cell lines (Appendix A) using ssGSEA. Hierarchical cluster analysis identified two distinct clusters, C1 and C2 (Figure 3A). C1 had high expression of BER and NER genes, while C2 was significantly upregulated in MMR and HR (*p* = 0.004) genes (Figure 3A). NHEJ, although appearing upregulated in C1 (Figure 3A), was not significantly altered between C1 and C2 (*p* = 0.154). To determine whether cell line DDR gene expression clusters had similarities to the TCGA gene expression clusters, hierarchical clustering was performed on combined ssGSEA scores of TCGA samples and cell lines. All cell lines from C1 clustered within a combined cluster resembling TC1, while all cell lines belonging to C2 did not associate with any TCGA DDR cluster (Appendix A).

To investigate the extent at which baseline gene expression in DDR pathways contributes to differential treatment response, cell viability of glioblastoma cells treated with clinically relevant doses of TMZ (35 μM) and/or RT (2Gy) was assessed (Figure 4). Across the cell lines, there was a differential response to TMZ, while most cell lines had similar sensitivity to single-dose RT (Figure 4). *MGMT* methylation status is a clinical biomarker of TMZ sensitivity [18]. As expected, *MGMT* methylated cell lines had a significantly reduced cell viability than *MGMT* unmethylated cell lines (*p* = 0.004) in response to TMZ (Appendix A). Three cell lines (HW1, FPW1 and SJH1) were identified as *TP53* mutants; however, their sensitivity to TMZ or RT was not significantly different compared to *TP53* wild-type cell lines (*p* = 0.39 and 0.86, respectively). Whilst distinct differences in DDR gene expression were observed between clusters C1 and C2, no significant differences in cell viability were observed in response to TMZ (*p* = 0.32) or RT (*p* = 0.097) (Figure 3B,C).

When stratified into “high” and “low” DDR expression groups, treatment response varied in certain pathways (Figure 3D,E). High gene expression of the NER pathway was associated with more TMZ resistance (*p* = 0.032), while cells with high MMR gene expression were more sensitive to TMZ (*p* = 0.0039) (Figure 3D). Interestingly, TMZ response did not significantly differ between high and low expression of BER, HR and NHEJ genes. When comparing RT response, high HR gene expression was associated with increased resistance to RT (*p* = 0.0038) whilst there was no significant difference in other DDR pathways (Figure 3E). Collectively, these results suggests that transcriptomic expression of NER and MMR pathways may influence TMZ sensitivity, while gene expression of the HR pathway may influence RT response.

### 3.3. Upregulation of DDR Genes after Standard Treatment in Glioblastoma Cell Lines

Few studies have specifically examined the expression of multiple DDR pathways in glioblastoma cells treated with TMZ or RT to determine the timing of treatment-induced DDR pathway activation and the extent at which they repair DNA. Here, we used qPCR to determine the expression of 84 DDR genes in glioblastoma cell lines (SB2b, FPW1, MN1 and HW1) treated with a clinically relevant dose of TMZ and/or RT at 2, 24 and 48 h post-treatment. These cell lines represented to various degrees different cluster groups, *MGMT* statuses, and responses to TMZ or RT treatment (Figure 3 and Figure 4). Within the context of this study, DEGs were identified as significantly up- or downregulated genes in treated cells compared to the untreated control at each specific time point.

When observing the frequency and distribution of DEGs across all cell lines, several trends appeared. Figure 5 summarises the accumulated degree of DDR upregulation and downregulation in all four cell lines treated with TMZ, RT, or TMZ + RT (Figure 5A), as well as the proportion of DEGs belonging to DDR pathways for the average cell line (Figure 5B–G). Furthermore, Figure 6 depicts the 16 most frequently significantly up- or downregulated genes across all cell lines, treatments, and time points. Notably, the majority of DEGs across all four cell lines were upregulated (88%). There appeared to be variability between cell lines and treatments (Appendix A); however, an overall trend across cell lines was the predominant upregulation of genes within 24 h after treatment, especially in RT- and TMZ + RT-treated cells (Figure 5A).

Differences in the frequency of DEGs were apparent between treatments across time points. On average, TMZ induced the lowest frequency of DEGs, whereas RT induced the most robust response causing the highest frequency of DEGs across each time point (Figure 5B,C). The combination treatment induced more DEGs than TMZ alone, but surprisingly less DEGs than RT alone (Figure 5D). This suggests that the addition of TMZ to RT-treated cells disrupts the dynamics of DDR in this context.

The frequency of DEGs belonging to specific DDR pathways was also distinct for each treatment between time points. Although variability in response was observed between cell lines (Appendix A), trends appeared when considering the average expression across all four cell lines. On average, cells treated with TMZ at 2 h had an upregulation of NER genes while several genes involved in BER, HR, NHEJ and MMR were upregulated 24 h after treatment (Figure 5B). Cells treated with RT had upregulation in all DDR pathways and remained consistent for the 2 and 24 h time points before a reduction in expression at 48 h (Figure 5C). For TMZ + RT-treated cells at 2 h, the BER, MMR, NER and HR pathways were upregulated, while NHEJ and NER genes increased in expression at 24 h post TMZ + RT treatment (Figure 5D). Then, at 48 h post TMZ + RT treatment, most genes returned to baseline levels where fewer genes were upregulated with NER expression the most prominent.

When considering the frequency of specific genes across all time points and treatments, genes from several pathways were represented, in particular HR and BER genes. *NEIL3* and *CCNO* (BER), *XRCC2*, *RAD54L* and *ATM* (HR), *DDB2* (NER), and *MSH2* (MMR) were among the most differentially expressed and upregulated (Figure 6). The NER gene, *ERCC8*, was the most frequently downregulated (*n* = 5) and appeared in cells treated with TMZ or TMZ + RT (Figure 6), suggesting that TMZ may influence its expression. From the three prognostically significant genes identified from the TCGA cohort, *RPA3* and *RAD51C* were upregulated only once across all cell lines while *ATP23* appeared to be upregulated in FPW1 cells and downregulated in the HW1 cell line (Appendix A). Overall, these results emphasise the broad response of DDR genes and pathways after DNA-damaging treatment.

## 4. Discussion

Glioblastoma is an extremely aggressive and treatment-resistant disease, often prone to recurrence and poor patient survival due to the failure of standard treatment. The activation of DDR pathways is a significant factor in reducing treatment efficacy, enabling efficient repair of treatment-induced DNA damage, and increasing the likelihood of tumour cell survival [5,6]. We investigated DDR through a transcriptional lens to identify whether DDR is a significant feature in glioblastoma survival and response to standard treatment.

Firstly, we identified three distinct clusters (TC1–3) of TCGA glioblastoma patients based on the overall expression of DDR pathway genes using ssGSEA. Despite the clusters displaying low, moderate, and high DDR gene expression, respectively, no significant OS differences were observed between clusters. A study by Meng et al. [19] found low DDR gene expression to indicate favourable prognosis in a combined cohort of low-grade glioma (LGG) and glioblastoma patients; however, consistent with our findings here, no survival difference was apparent in glioblastoma patients alone. Interestingly, a trend appeared, whereby the proportion of *TP53* alterations increased from TC1 to TC3 and thus aligned with the extent of DDR expression across each cluster. This may be the case as *TP53* alterations can enhance genomic stress within rapidly dividing cells and thus induce an increased activation of DDR to counteract such stresses [20]. Furthermore, a similar trend occurred for *MGMT* methylation, where the proportion of *MGMT* methylated patients increased from TC1 to TC3. Given that *MGMT* methylation plays a significant prognostic role in determining a longer overall survival in glioblastoma patients [18], its higher proportion within TC3, together with low samples sizes, may play a factor as to why no significant survival was observed even though higher DDR expression was evident. Despite this, we investigated individual DDR gene expression and their influence on OS outcomes. From the 84 DDR genes assessed, the high expression of *ATP23*, *RAD51C* and *RPA3* was independently associated with poor OS outcomes in the TCGA IDH wild-type glioblastoma cohort. All three genes play roles in the repair of DSBs and may enhance treatment resistance. For instance, *ATP23*, a commonly amplified gene within glioblastoma, is involved in NHEJ and is upregulated in response to RT [21]. *RAD51C* plays an important role as a stabiliser of complexes involved in HR [22], while *RPA3* is part of the three-subunit replication protein A (RPA) complex involved in HR and DSB repair and has been implicated in glioblastoma OS outcome [23]. These data suggest that DDR gene expression influences patient outcome and warrants further investigation on the role DDR plays in glioblastoma treatment resistance.

Stringer et al. [13] described efforts to fully characterise the 12 patient-derived glioblastoma cell lines, including use in a xenograft model to show they are morphologically representative of the patient’s original tumour. This would suggest that the cell lines used in the current study are a good representation of the original tumour. We investigated the DDR baseline gene expression in these cell lines and found two distinct clusters (C1 and C2) with differential pathway expression. When compared to the glioblastoma TCGA cohort, four cell lines in the C1 cluster aligned with TC1 cluster of the TCGA cohort with respect to baseline DDR gene expression. The TCGA cohort was a significantly larger sample size at 140 cases, and thus it was surprising that 8 of the cell lines did not align with a TCGA cluster in baseline DDR gene expression. Furthermore, the C1 and C2 cell line clusters do not appear to have any relationship to known characteristics of glioblastoma such as *MGMT* methylation status. In regard to these discrepancies between the TCGA data and the baseline data from the 12 patient-derived glioblastoma cell lines, Stringer et al. showed in their original publication of the cell line data that 7 of the 12 cell lines maintained a molecular signature equivalent to that from the original tumour tissue [13]. As such, we cannot exclude the possibility that the cell culture conditions affected the baseline gene expression signatures [24]. However, with respect to the cell lines treated with TMZ and/or RT, our analysis used a non-treated control which will negate the effect of factors such as cell culture conditions, as the only change between groups is the treatments. With respect to the aim of investigating DDR gene expression and effects on treatment response, there was no difference in response to either RT or TMZ between the two cell line clusters, suggesting that the cell lines were responding in a similar manner to the treatment despite a difference in baseline DDR gene expression. Further investigation of the characteristics of the glioblastoma cell lines may help to explain the differences in the C1 and C2 DDR clusters.

When examining individual pathway expression in the cell lines, we showed that genes in DDR pathways are associated with response to TMZ or RT. Cells with a higher expression of NER genes were more resistant to TMZ, while high expression of MMR genes conferred TMZ sensitivity. Previous studies have suggested similar trends with respect to NER components [8,25], while alterations in MMR genes such as MSH6 have been associated with TMZ resistance and recurrent glioblastoma tumours [26]. Our data also showed that gene expression of the HR pathway was inversely associated with RT response. The inhibition of HR components has enhanced radiosensitivity in glioblastoma cells [27,28], hence cells with a higher expression of HR genes may be more likely to survive than tumours which have a lower baseline expression [4,6,17,29]. Further investigation is needed to explore these results in more depth.

A limitation to this analysis is the bulk transcriptomic lens of tumours prior to any treatment, which does not adequately reflect changes in DDR expression from standard treatment. Thus, studying DDR across a pre- and post-treatment time course may reveal greater insight. In this regard, one of the important aspects of the work presented here is the time course analysis of DDR gene expression response, showing that several genes and pathways are upregulated in response to standard treatment, especially RT. These data may inform the feasibility of targeting DDR components to enhance treatment response in glioblastoma patients. One such approach is being explored through the development of small-molecule inhibitors of DDR proteins including poly (ADP-ribose) polymerase (PARP), Wee1, checkpoint kinase 1 and 2 (Chk1 and Chk2), ataxia-telangiectasia mutated (*ATM*), ataxia-telangiectasia and rad3-related (ATR), and DNA-dependent protein kinase (DNA-PK) [4,30]. Across all treatments, DDR gene expression changes occurred within a 24 h period after treatment. RT had the greatest response, with numerous pathways showing gene upregulation, suggesting activation at 2 and 24 h after treatment. TMZ on the other hand had a lower level of DDR gene upregulation. The composition of genes was also different to RT, with a predominance of NER genes expressed at 2 h before a sharp increase in BER gene upregulation, as well as upregulation in HR, MMR and NHEJ genes at 24 h. These observations agree with the understanding that O^6^-meG adducts caused by TMZ form DNA breaks only after several replication cycles have occurred [31,32], in contrast to RT, which results in immediate DNA damage formation [29]. The combination of TMZ and RT resulted in a comparative decrease in the number of differentially expressed DDR genes compared to RT alone across all time points, and the up- or downregulation of DDR genes from either RT or TMZ alone was not always observed in the cells treated with TMZ + RT. This may have occurred as the two treatments alone produce varying degrees of DDR activation across different time points and pathways.

Across the four cell lines, several DDR genes were frequently observed as differentially expressed. The most notable and frequent included *NEIL3, XRCC2, CCNO, RAD54L, ATM*, *DDB2,* and *MSH2*, all of which have been linked to treatment resistance in glioma or solid tumours such as colorectal cancer [10,26,33,34,35,36,37,38]. Interestingly, the expression of *RPA3* and *RAD51C* rarely changed although being associated with OS outcomes in the TCGA cohort, while *ATP23*, was either upregulated or downregulated depending on the cell line. This suggests that baseline expression and associated patient outcome may not entirely capture the direct role such genes play in DDR when examined across time, and further investigation is needed. This study also shows the upregulation of several NER genes in response to TMZ and RT, which is an underreported DDR pathway in glioblastoma. ATR, a global sensor of DNA damage, has been implicated in NER activation [39] and is thus a potential therapeutic target. The transcriptional lens of this study, however, cannot conclusively answer this and an in-depth analysis is required to elucidate the exact function that NER and its components play in response to the standard treatment of glioblastoma, which may reveal druggable targets for its inhibition. Furthermore, our study of DDR expression has focused on characteristic DDR genes with less emphasis on DNA polymerases and ligases that play overlapping functions across pathways [40,41,42]. Thus, future work will seek to include these genes with overlapping functions to gain greater insight into the response of DDR pathways and their influence on treatment resistance.

Overall, this study reveals an influence of DDR genes and subsequent pathways in glioblastoma cellular responses to treatment. Specific clusters of DDR expression failed to show significant differences in patient survival outcome or cell line response to TMZ or RT. However, our analysis revealed that the high expression of three DDR genes associated with poorer overall patient survival, while expression of MMR, NER and HR influenced sensitivity to TMZ or RT in glioblastoma cell lines. Our results suggest that the DDR is primarily upregulated within a 24 h period after treatment of TMZ and/or RT, with distinct trends of DDR activation apparent between treatments. Such data give insight into the changes in DDR gene expression in response to standard treatment and the potential for targeting commonly upregulated DDR components to produce radio- or chemo-sensitising agents.

## Figures and Tables

**Figure 1 cells-11-01215-f001:**
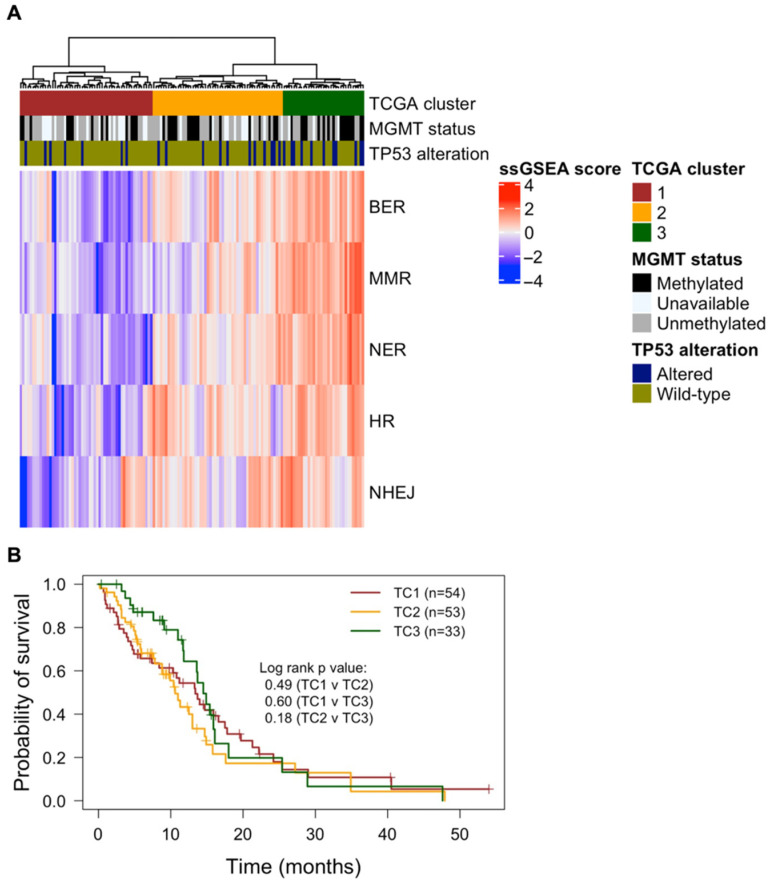
Transcriptomic profile of DDR pathways of glioblastoma patients in the TCGA cohort (*n* = 140). (**A**) Log-transformed ssGSEA scores are represented for each DDR pathway; and after hierarchical clustering (Ward’s method), three distinct TCGA clusters were identified (TC1, TC2, and TC3). The Kruskal–Wallis test was used to compare ssGSEA scores of each pathway between clusters, with the same trend followed across all DDR pathways: TC1 < TC2 < TC3 (*p* < 0.01). *MGMT* methylation status and *TP53* alterations (SNVs or homozygous deletions) are also depicted for respective patient samples. (**B**) Kaplan–Meier plots of OS are shown for patients within DDR clusters (TC1—red; TC2—yellow; TC3—green). The log-rank test was performed between each combination of clusters, revealing no significant OS differences between clusters. Statistical significance was determined with a *p*-value < 0.05.

**Figure 2 cells-11-01215-f002:**
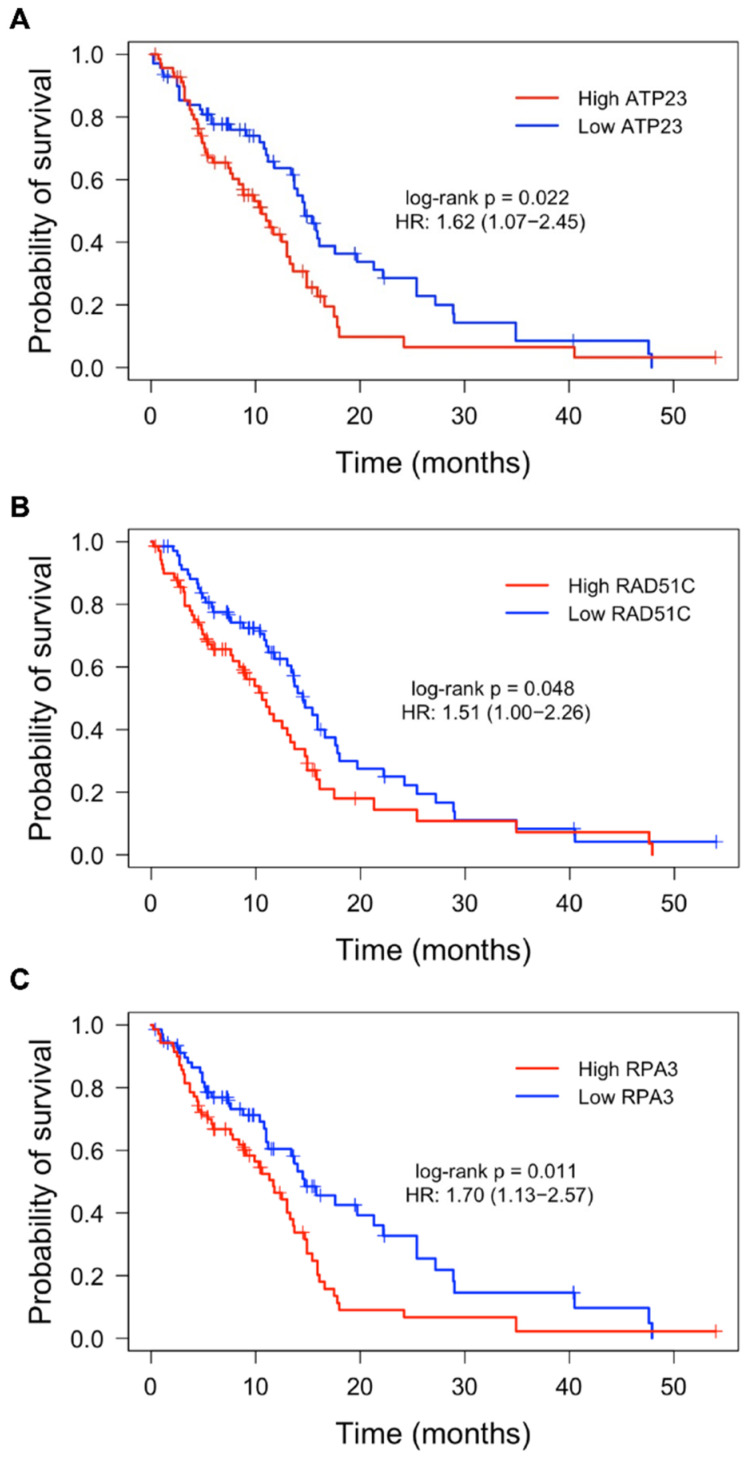
Kaplan–Meier plots of OS for high and low expression of *ATP23*, *RAD51C* and *RPA3* in the TCGA glioblastoma cohort. Patients were stratified into “high” (red) and “low” (blue) expression groups based on a median split of RNA expression for each respective DDR gene. Across all samples (*n* = 140), high expression of *ATP23* (**A**), *RAD51C* (**B**) and *RPA3* (**C**) was significantly associated with lower OS. Log-rank *p*-values and hazard ratio prior to multiple Cox regression are shown. Significance was established in genes with *p*-value < 0.05 after both a log-rank test and multiple Cox regression of significant clinical features (i.e., therapy).

**Figure 3 cells-11-01215-f003:**
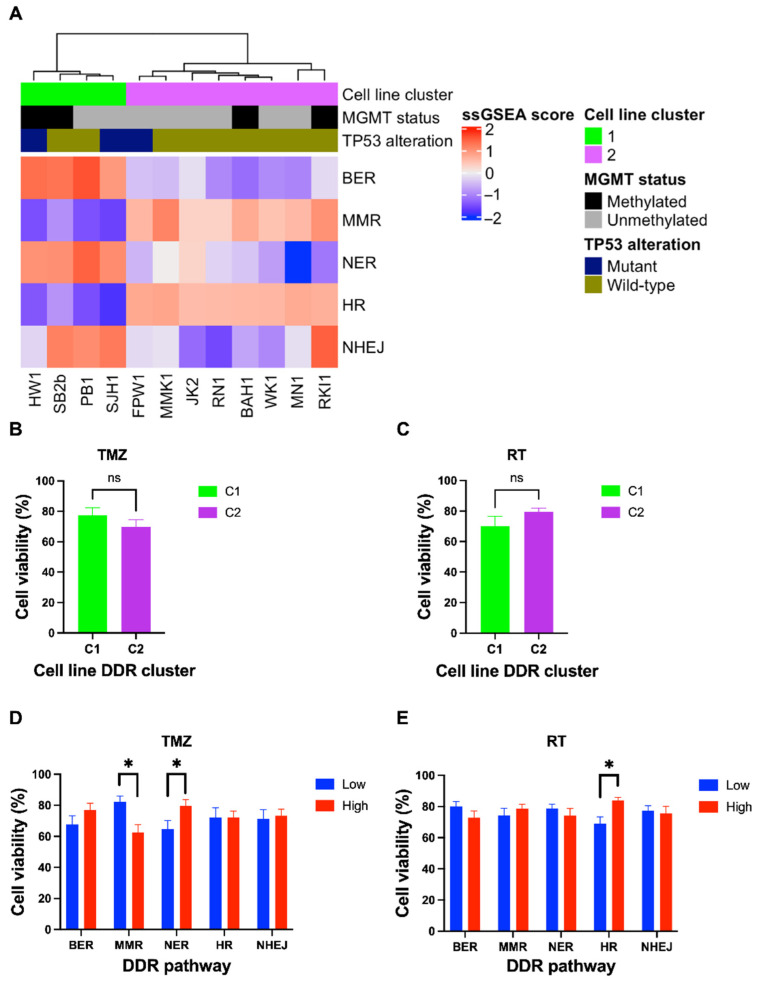
Transcriptomic profiling of DDR pathways in 12 patient-derived glioblastoma cell lines and response to standard treatment. (**A**) Log-transformed ssGSEA z-scores are shown corresponding to relevant DDR pathways, including *MGMT* methylation status and *TP53* mutation of glioblastoma cell lines. Two distinct clusters were identified (C1 and C2), in which C1 had a significant upregulation of BER and NER pathways (*p* = 0.004), while MMR and HR were upregulated in C2 (*p* = 0.004). Cell lines were treated with a clinically relevant dose of TMZ (35 µM) or RT (2 Gy) for 7 days, cell viability assessed using MTT assay. Cell lines were grouped and compared using a student’s *t*-test to assess differences in TMZ and RT response based on DDR cluster (**B**,**C**) as well as “high” and “low” expression of DDR pathways (**D**,**E**). There was no significant difference in cell viability between the C1 or C2 cell line clusters after TMZ or RT treatment (**B**,**C**). TMZ sensitivity was associated with high MMR gene expression, while TMZ resistance was associated with high NER gene expression (**D**). RT resistance was associated with high HR gene expression (**E**). *p*-values < 0.05 were considered significant (* *p* < 0.05) (ns = non-significant).

**Figure 4 cells-11-01215-f004:**
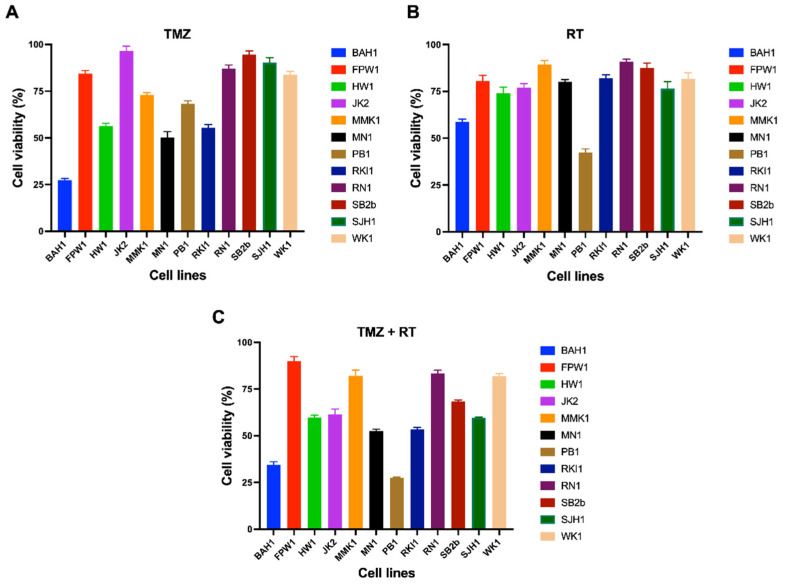
Cell viability of 12 patient-derived glioblastoma cell lines treated with TMZ (**A**), RT (**B**) or TMZ + RT (**C**). Cell lines were treated with a clinically relevant dose of TMZ (35 µM) and/or RT (2 Gy), grown as adherent cultures for 7 days before being assessed for cell viability with an MTT assay. Data points represent the mean and SEM of three biological replicates over three independent experiments.

**Figure 5 cells-11-01215-f005:**
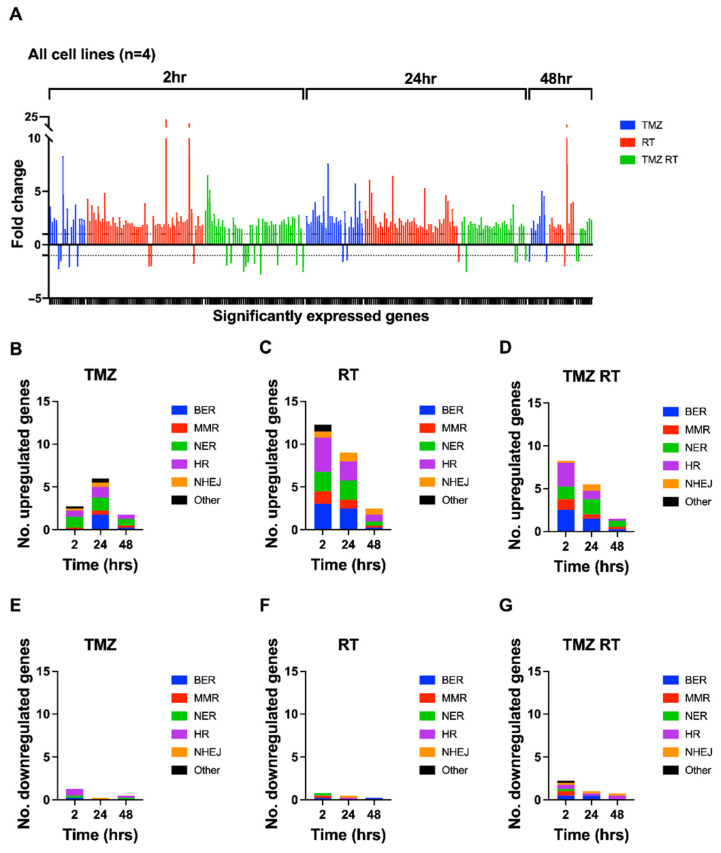
Accumulated frequency and distribution of DEGs across four patient-derived glioblastoma cell lines (FPW1, HW1, SB2b and MN1) in response to treatment. Cell lines were treated with RT (2Gy) and/or TMZ (35µM) before being harvested for RNA extraction at 2, 24 and 48 h after treatment. Quantitative PCR using a Custom TaqMan array card was undertaken to assess mRNA expression of 84 DDR genes. (**A**) Fold changes (±SEM) represent the sum total of DEGs across all cell lines, treatments (TMZ- blue; RT—red; TMZ + RT—green), and time points. Positive fold-changes (>1) represent upregulated genes and negative fold-changes (<−1) represent downregulated genes, while the area between 1 and −1 represents baseline expression. To view genes and their fold-changes in order of (**A**), see Appendix A. (**B**–**G**) The average numbers of upregulated or downregulated genes across all cell lines for TMZ- (**B**,**E**), RT- (**C**,**F**) and TMZ + RT- (**D**,**G**) treated cells are graphically shown and represent the proportion of pathways that the DDR genes belong to at given time points after treatment. DDR pathways for each gene are depicted in Appendix A. BER—blue; MMR—red; NER—green; HR—purple; NHEJ—orange; other—black.

**Figure 6 cells-11-01215-f006:**
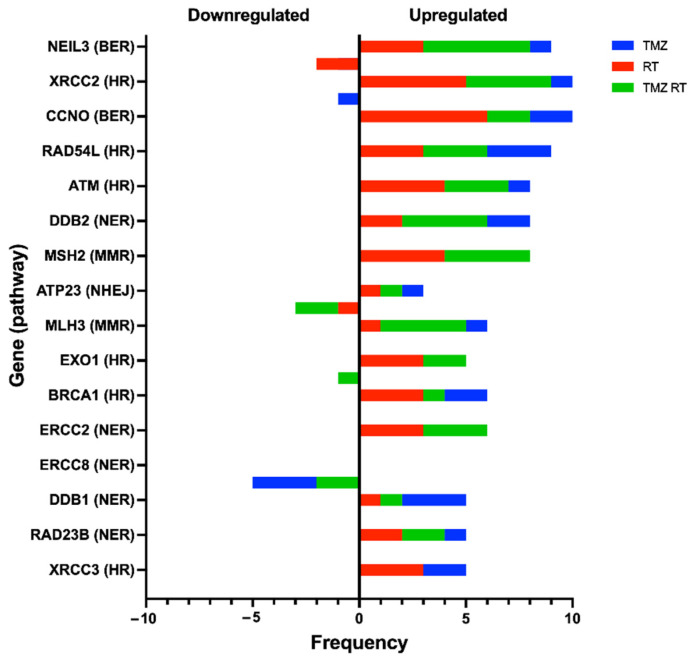
Top 16 most frequently up- or downregulated genes across all four glioblastoma cell lines (FPW1, HW1, SB2b and MN1). Data include the accumulated frequency across all variables for each respective gene, including the number of upregulated (positive values) or downregulated (negative values) occurrences, and the occurrence of this change in each specific treatment (TMZ = blue, RT = red, TMZ RT = green).

## Data Availability

Research data are stored in an institutional repository and will be shared upon request to the corresponding author.

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
