# Peer review of "Transcriptomic Profiling of DNA Damage Response in Patient-Derived Glioblastoma Cells before and after Radiation and Temozolomide Treatment"

_cells, 2022, doi:10.3390/cells11071215_

Round 1
Reviewer 1 Report
The authors have conducted an extensive transcriptome analysis to address the association between glioblastoma (GBM) senstitivity to clinically relevant doses of radiation (RT), temozolomide (TMZ) or combined RT+TMZ and DNA damage response (DDR) pathways or individual DDR genes. Two types of gene expression data were used - the TCGA dataset derived from the transcriptome profiling of whole-tissue GBM samples and experimental dataset generated in this study using a panel of 12 GBM cell lines. The results obtained from both datasets support the overall role of DDR in determining GBM sensitivity to RT and TMZ and indicate that these treatments migh impact diffferent DDR pathways in GBM.
Analyses of the TCGA dataset have revealed that high expression of three DDR genes (ATP23, RAD51C, RPA3) is independently associated with poorer overal survival. This finding is novel and has potential clinical relevance. Less clear is the relevance of results obtained with cultured GBM lines. One of the major concerns is that GBM lines used in transcriptome analyses have been cultured in the presence of high concentrations of penicillin and streptomycin, antibiotics that are known to induce systemic changes in gene expression in human cell lines and exert broad-spectrum effects at the gene regulatory level. In light of this concern, the impact of culture-related conditions on transcriptomic changes induced by RT or TMZ in GBM cell lines cannot be ruled out. This could also explain the discrepancy of results obtained with TCGA data and cultured GBM lines.
Statement that "all glioblastomas have the IDH1-wild type gene" is misleading as IDH1 is mutated in more than half of secondary GBMs.
Reviewer 2 Report
In the manuscript „Transcriptomic profiling of DNA damage response in patient-derived glioblastoma cells before and after radiation and temozolomide treatment” M. Lozinsky and colleagues seek for the correlation between DDR (DNA damage response) gene expression levels and overall survival of TCGA GBM patients cohort and sensitivity of GBM cell lines to genotoxic insults. Moreover, they assess DDR gene expression in response to DNA damage. DNA damage sensors and proteins playing parts in various DNA repair modules build large and intertwined system, regulated at the transcriptional and posttranscriptional levels, depending on mutational (e.g. TP53, BRCA1) and epigenetic (e.g. MGMT & TERT promoters) tumor/cancer cell status, gene amplification or deletion etc. Therefore, taking into account multilevel heterogeneity of GBM tumors, it’s very challenging to find specific gene expression pattern predicting patient/cells response to DNA damaging agents analyzing low number of samples.
The in vitro experiments e.g. cell treatment with use of relevant dose of TMZ and RT for 7 days are reasonably planned. Frequently, biologists apply much higher amount/intensity of TMZ/RT, impossible to be obtained in vivo.
Gene selection (Table S1) used for subsequent analyses do not take into account that some genes (like ligases or DNA polymerases) overlap and are necessary for different DDR systems function. Now it looks like the most characteristic, “unique” genes are chosen. Although understandable from technical point of view, this approach may be suboptimal for the thorough analysis of each DDR involvement.
There are some minor points to address:
- Around 30% of GBM patients harbor mutation(s) in TP53, many of them were shown to be/are expected to be oncogenic. Does TP53 status affect DDR expression pattern in TCGA patients group, especially DDR genes known to be regulated by TP53?
- What is the effects of TP53 mutations observed in cell lines on TP53 function? What was the rationale behind choosing cell lines for gene expression experiments?
- It would be better for the paper flow if data from supplementary Figure S2 are moved to the main figures. Moreover, the figure should be supplemented with the combined effect of TMZ and RT on cell viability.
- The letters at X axis (gene names?) in Figure 4A are not visible.
- It would be more informative to put cell line names below heatmap in Fig 3A, instead of showing the status of particular DDR for every cell line in Table 1.
Round 2
Reviewer 1 Report
the authors have addressed previous concerns raised during the first round of revision